# Novel Platform for Regulation of Extracellular Vesicles and Metabolites Secretion from Cells Using a Multi-Linkable Horizontal Co-Culture Plate

**DOI:** 10.3390/mi12111431

**Published:** 2021-11-21

**Authors:** Takeo Shimasaki, Satoko Yamamoto, Risa Omura, Kagenori Ito, Yumiko Nishide, Hideki Yamada, Kazumi Ohtomo, Tomo Ishisaka, Keiichiro Okano, Takenori Ogawa, Hiroyuki Tsuji, Yoichi Matsuo, Toshinari Minamoto, Naohisa Tomosugi, Etienne Ferain, Takahiro Ochiya

**Affiliations:** 1Medical Research Institute, Kanazawa Medical University, Uchinada 920-0293, Japan; s-yama@kanazawa-med.ac.jp (S.Y.); r-omura@kanazawa-med.ac.jp (R.O.); tomosugi@kanazawa-med.ac.jp (N.T.); 2Division of Translational and Clinical Oncology, Cancer Research Institute, Kanazawa University, Kanazawa 920-0934, Japan; minamoto@staff.kanazawa-u.ac.jp; 3Ginrei Lab Inc., Uchinada 920-0293, Japan; 4Department of Urology, Jikei University School of Medicine, Tokyo 105-0003, Japan; ito.kagenori@gmail.com; 5Division of Cellular Signaling, National Cancer Center Research Institute, Tokyo 104-0045, Japan; tochiya@tokyo-med.ac.jp; 6Shinko Chemical Co., Ltd., Kanazawa 920-0346, Japan; master@shinko-ccl.jp (Y.N.); hyamada@shinko-ccl.jp (H.Y.); ohtomo@shinko-ccl.jp (K.O.); 7Department of Head and Neck Surgery, Kanazawa Medical University, Uchinada 920-0293, Japan; tomo-n@kanazawa-med.ac.jp (T.I.); okano-k@kanazawa-med.ac.jp (K.O.); tsujih-i@kanazawa-med.ac.jp (H.T.); 8Department of Otolaryngology-Head and Neck Surgery, Gifu University Graduate School of Medicine, Gifu 501-1194, Japan; ogawa@gifu-u.ac.jp; 9Department of Gastroenterological Surgery, Nagoya City University Graduate School of Medical Sciences, Nagoya 467-8601, Japan; matsuo@med.nagoya-cu.ac.jp; 10it4ip S.A., 1348 Louvain-la-Neuve, Belgium; ferain@it4ip.be; 11Department of Molecular and Cellular Medicine, Institute of Medical Science, Tokyo Medical University, Tokyo 160-0023, Japan

**Keywords:** co-culture plate, multiple connection, horizontal co-culture, extracellular vesicles, exosome, track-etched-membrane, metabolite

## Abstract

Microfluidics is applied in biotechnology research via the creation of microfluidic channels and reaction vessels. Filters are considered to be able to simulate microfluidics. A typical example is the cell culture insert, which comprises two vessels connected by a filter. Cell culture inserts have been used for years to study cell-to-cell communication. These systems generally have a bucket-in-bucket structure and are hereafter referred to as a vertical-type co-culture plate (VTCP). However, VTCPs have several disadvantages, such as the inability to simultaneously observe samples in both containers and the inability of cell-to-cell communication through the filters at high cell densities. In this study, we developed a novel horizontal-type co-culture plate (HTCP) to overcome these disadvantages and confirm its performance. In addition, we clarified the migration characteristics of substances secreted from cells in horizontal co-culture vessels. It is generally assumed that less material is exchanged between the horizontal vessels. However, the extracellular vesicle (EV) transfer was found to be twice as high when using HTCP. Other merits include control of the degree of co-culture via the placement of cells. We believe that this novel HTCP container will facilitate research on cell-to-cell communication in various fields.

## 1. Introduction

Systems that utilize microstructures, such as microfluidic channels, are being increasingly applied in biotechnology research. Cell culture inserts, which are used for the co-culture of cells, have utilized microstructures for many years. The trademarked system “Transwell^®^” is a well-known co-culture vessel that comprises two containers that are stacked vertically. The upper container is composed of a filter at the bottom through which the substances in the two containers can interact. This structure is essentially the same as a microfluidic machine because micro-materials that are secreted by the cells pass through a filter or microfluidic channel to affect distant cells. Thus, co-culture research is one of the most relevant fields in which microstructures have been applied.

Recently, several co-culture methods have emerged, and they can be broadly classified into two types [1]: (1) direct co-culture, in which cells are mixed in a single well without separation, and (2) indirect co-culture, in which two different cell types are cultured simultaneously and isolated from each other using a separation mechanism. Although direct co-culture is more commonly employed, the use of this approach to analyze individual cells is complicated. In contrast, indirect co-culture involves placing the cells in different environments, and cell–cell interactions are mediated via humoral factors, meaning that it is easier to observe the cells because they are not mixed. Indirect co-culture systems are mainly classified into three types: (1) separation by filters, (2) separation by a semi-solid material such as a gel, with humoral factors exchanged via the gel, and (3) the formation of separate cell colonies or layers by culturing, in which complete isolation of cell types is not possible [1]). A typical example of filter separation is a cell culture insert, which is characterized by a bucket-in-a-bucket structure that is constructed vertically. This type of co-culture plate is therefore referred to as a vertical-type co-culture plate (VTCP).

The most popular type of VTCP is the Boyden chamber, which was developed by Stephan Boyden in 1962 [2]. It was initially designed to study cell invasion by counting the number of cells that passed through a filter [3]. In the Boyden chamber co-culture method, the culture fluids at the top (on the insert) and the bottom (on the bottom dish) of a system are shared for cell–cell interaction studies. This type of VTCP is generally referred to as a “Transwell” style [4,5,6] or “cell culture insert” type [7].

VTCPs have long been used as the standard for co-culture, and this technique has not changed fundamentally because of the difficulty in making novel vessels. When using this design, it is expected that cells will grow on an insert, and that the liquid factors will be exchanged with the cells in the lower wells through the filter. However, this is not always achieved, mainly because of the fundamental problems associated with high cell densities. There are three main issues when using VTCPs:The top container is invisible: Neither cell line could be simultaneously observed under a microscope. The cells in the culture vessel are usually observed from the bottom of the vessel because the culture fluid hinders observation from the top of the vessel. The upper container cannot be observed because of the short focal length of the microscope. Therefore, cells were only observed in the lower parts of the VTCP-type vessels (Appendix A).Different materials could impact the results: The cells are not cultured on the same type of material surface: one group of cells is cultured on the filter material and the other on a plastic material; this difference could significantly impact the experimental results (Appendix A).High cell densities are not considered: A high density of cells may prevent co-culture due to blockage of the filter. As the Boyden chamber was initially invented to evaluate cell invasion [2,8], scenarios in which cells were present at high densities were not considered (Appendix A).

To overcome these problems and facilitate studies on cell–cell communication, we developed a novel horizontal-type co-culture plate (HTCP), as shown in Figure 1. This design is based on recent advances in three-dimensional computer-aided design technology, which offers novel possibilities for creating different culture vessels. We hypothesized that a horizontal system would solve the problems associated with VTCPs; it would enable simultaneous observation of cells in both containers using an inverted microscope, through which a variety of cellular interactions and cell morphologies during the maturation of each cell line can be observed.

In addition, researchers can freely choose from a variety of commercially available filters that are 13 mm in diameter to control the sharing factors of the liquid media when using HTCP, meaning that a wide variety of filter options can be used with the system.

Intercellular communication has traditionally been considered to involve the interaction of cells with their neighbors through direct interactions that are mediated by soluble molecules, such as cell-secreted cytokines. However, recent studies have revealed that Extracellular vesicles (EVs) play an important role in cell–cell communication [9,10]. Johnstone et al. [11] first discovered vesicles secreted by sheep reticulocytes [12,13] in 1987, which were subsequently named exosomes. Exosomes are a type of EVs that are secreted by many cells and are present in body fluids, including blood, urine, and saliva. Although the detailed features of exosomes remain unknown, the discovery of microRNA and RNA in EVs in 2006–2007 demonstrated the possibility that informative genetic transmission between cells may occur via exosomes [14,15]. Since then, many researchers have turned their attention to EVs, and exosomes have since become a hot topic of study in various fields. EVs also include slightly larger plasma membrane-derived microvesicles and apoptotic bodies that are released during cell apoptosis [16]. Although the lack of a clear definition has led to confusion among researchers, the description of an exosome as an EV has been driven mainly by the International Society for Extracellular Vesicles [17]. EVs are currently divided into three main groups: (1) exosomes that are 50–150 nm in diameter and are bound by lipid bilayer membranes, (2) microvesicles that are 100–1000 nm in size and are directly secreted by cells, and (3) apoptotic bodies that arise from cell apoptosis [16]. Since the precise definition of exosomes remains elusive because of the lack of standard protocols for identifying and distinguishing exosomes and microvesicles, exosomes have recently begun to be referred to as EVs. Although exosomes are released via exocytosis and microvesicles are released via plasma membrane shedding, EVs are an appropriate term to describe both because the different types of vesicles cannot be distinguished by ultracentrifugation.

As it became clear that extracellular vesicles play an essential role in communication between cells, it became necessary to analyze them separately. The EVs can be separated from cells using three main methods [18]: (1) the ultracentrifugation method that uses weight differences, (2) using the properties of the surface proteins on extracellular vesicles, and (3) using the difference in the sizes of the EVs. The size-separation method is the most suitable method for use with microfluidic devices and requires either a filter or the fabrication of tiny pores in the device to enable size separation. The problem with this method is that extracellular vesicles are extremely small, with exosomes reaching only 50 to 150 nm in size. Track-etching technology is ideally suited for producing filters with uniform pores. Track-etched membrane filters were fabricated by irradiating a raw polymer film with high-energy heavy ions, creating damaged linear tracks across the film, which were later converted into pores by selective wet chemical etching (Appendix A). Etching conditions can be adjusted to obtain uniform pores with a precise diameter adapted to separate the extracellular vesicles (Appendix A). In addition, for the efficient filter, it is necessary to prevent substances from leaking across the pores in the filter. VTCPs are made by molding the culture vessel and filtering together, so that the filter cannot be replaced. In the HTCP, an O-ring is used to sandwich the filters, meaning that it is possible to replace the filter. We are the first group to our knowledge, to commercialize containers with multiple horizontal connections. HTCP has been reported in several studies [19,20]. However, the migration characteristics of substances secreted by HTCP cells have not yet been reported. The novel HTCP vessel is expected to help advance cell co-culture research, including shedding new light on the role of EVs as messengers that can deliver signals between cells. Here, we describe the design of a HTCP system as a new type of co-culture vessel. We also describe the results of experiments conducted to evaluate the feasibility of exchanging liquid factors using a track-etched membrane in comparison to that of using a VTCP. The functional ability of the track-etched membrane to separate EVs was examined to elucidate the influence of EVs on co-cultures.

## 2. Materials and Methods

### 2.1. Device Design and Fabrication

The design of the HTCP, its assembly, size information, and how to use it are depicted in Figure 2 and Appendix A. All components comprising the HTCP were designed using Rhinoceros 6. Available online: https://www.rhino3d.co.jp/ (accessed on 14 November 2021). The products were prepared via injection molding by Shinko Chemical Co., Ltd. The design aims to achieve an easy transition from a monoculture to co-cultures, with or without an optional filter, and the option to culture separately with the possibility of transitioning to the co-culture mode at any time.

This product (HTCP) is already commercially available through AR Brown Co., Ltd. (Tokyo, Japan), Fujifilm Wako Pure Chemicals Corporation (Tokyo, Japan), Blast Inc. (Kawasaki, Japan), AdvantigenBioScience L.L.C. (Madison, WI, USA), Bulldog Bio. Inc. (Portsmouth, NH, USA), Xceltis GmbH (Mannheim, Germany), and 1st PhileKorea Inc. (Seoul, Korea). The product is named “ICCP”, “NICO-1”, “UniWells”, and “Pair-N-Share Tandem Co-Culture Wells & Plates”, but the product itself is the same.

### 2.2. Cell Lines

Human pancreatic cancer PANC-1 and AsPC-1 cells were obtained from RIKEN BRC (Tsukuba City, Japan). The cell lines were passaged for fewer than six months after resuscitation, cultured in complete Dulbecco’s modified Eagle medium (DMEM)-high glucose with 1% L-glutamine, 1% sodium pyruvate, 10% fetal bovine serum, and 1% penicillin-streptomycin (FUJIFILM Wako Pure Chemical Corporation, Osaka, Japan), and maintained at 37 °C with 5% CO_2_ in DMEM. To establish stable pancreatic cancer cell lines expressing CD63-GFP (gPANC1) and CD63-RFP (rPANC1), cells were transfected with the lentiviral vectors pCT-CD63-GFP and pCT-CD63-RFP (pCMsV, exosome/secretory, CD63 tetraspanin tag, System Biosciences, CYTO120-PA-1, CYTO120R-PA-1), respectively, as reported previously [21]. Supernatants containing lentiviral particles obtained from the gPANC1 cell line and transduced with pCT-CD63-GFP were filtered through a 0.6-μm filter and then applied for EV isolation.

### 2.3. Microscopy

Microscopic images were acquired using the EVOS system (Thermo Fisher Inc., Waltham, MA, USA) to confirm cell conditions and obtain fluorescent images. Excitation wavelengths of 488 nm and 561 nm were used to detect GFP fluorescence (gPANC1) and RFP fluorescence (rPANC1), respectively.

### 2.4. Cell Viability

According to the manufacturer’s instructions, cell viability was analyzed using the Superoxide Dismutase (SOD) Assay Kit-WST (FUJIFILM WAKO Pure Chemical Corporation, Osaka, Japan. S311). In brief, WST reagent (10 µL) was added to each well, and the sample was incubated for 45 min at 37 °C. Absorbance was measured at 450 nm (A450) and 620 nm (A620). The value obtained by subtracting A620 from A450 was used as the viable cell number.

### 2.5. Nanoparticle Tracking Analysis

Nanoparticle tracking analysis was used to visualize and measure the size of small particles (10–1000 nm) in the suspension based on Brownian motion analysis from a video sequence. We used the NanoSight system (NS300, Malvern Instruments Ltd., Malvern, UK) to detect EVs. The measurements were performed at least thrice; the mean value and standard error were calculated and are represented in the graph.

### 2.6. EV Isolation

EVs were isolated using the MagCapture™ Exosome Isolation Kit PS 293-77601 (Fujifilm Wako Pure Chemical Corporation, Osaka, Japan) as described previously [22].

### 2.7. Whole-Cell Protein Extracts

Cells were lysed with radioimmunoprecipitation (RIPA) buffer (FUJIFILM Wako Pure Chemical Corporation, Osaka, Japan) composed of 50 mM Tris-HCl (pH 7.4), 1% NP-40, 0.5% sodium deoxycholate (SDC), 0.1% sodium dodecyl sulfate (SDS), 150 mM NaCl, and a protease inhibitor cocktail (FUJIFILM WAKO Pure Chemical Corporation, Osaka, Japan). Supernatants obtained after centrifugation at 15,000× *g* for 10 min at 4 °C were used as whole-cell proteins.

### 2.8. Measurement of Glucose, Lactate, and Ammonium Ion Concentrations

The culture supernatant was filtered at each growth phase through a 0.22 μm membrane and directly analyzed with a BioProfile FLEX2 analyzer following the manufacturer’s instructions (Nova Biomedical, Inc., Waltham, MA, USA). The measurements were performed at least thrice; the mean value and standard error were calculated and are represented in the graph.

### 2.9. Protein Assay and Western Blotting

The protein was resuspended in 5× RIPA buffer (125 mM Tris-HCl, 750 nM NaCl, 5% NP-40, 5% SDC, and 0.5% SDS), after which the suspension was sonicated for 5 min and incubated for 15 min on ice. Protein extract concentrations were measured using Coomassie Protein Assay Reagents (Pierce, Thermo Fisher Scientific, Waltham, MA, USA). A 20 μg aliquot of protein was separated by SDS-polyacrylamide gel electrophoresis and analyzed by Western blotting to detect the proteins with anti-flotillin-1 (Cell Signaling Technology 13174), anti-CD-9 (CosmoBio Japan SHI-EXO-M01), anti-CD-81 (CosmoBio Japan, SHI-EXO-M03), and anti-CD-63 (BD-BioScience, Franklin Lakes, NJ, USA, 556019), followed by incubation with the secondary antibodies goat anti-mouse IgG (Thermo Fisher Inc., #31430), and anti-rabbit IgG (Thermo Fisher Inc., #31460). Electroblotted membranes (Amersham) were blocked with 5% bovine serum albumin (BSA). The expression of β-actin was used as a reference for the amount of protein in each cell sample. The signals were measured using enhanced chemiluminescence.

### 2.10. Silver Staining

The volume of proteins that passed through the filter was determined using a silver staining kit (FUJIFILM Wako Pure Chemical Inc., Osaka, Japan) according to the manufacturer’s instructions.

## 3. Results

### 3.1. Device Design and Description

The HTCP consists of four main parts: the top cover, O-ring, sidewall cover, and Body A or Body B, as detailed in Figure 2. Along with these, either a stand-alone structure for separate culture (Figure 2a) or a connected vessel that allows the same medium to be shared by different cell types through the filter (Figure 2b) can be established for co-culture. Multiple connections are also possible by using a Body C structure, which has a coupling mechanism on both the left and right sides of the container (Figure 2c), and it can allow three, four, or multiple connections. The stand-alone mode is used for culturing under different conditions, such as high oxygen and low oxygen, high temperature and low temperature, or with different chemical materials. Each well is designed to hold 1.8 mL of culture medium. The outer dimensions with slide glass size adaptors are the same as those of a standard microscope slide glass, allowing for simple time-lapse microscopic observations. We also designed a 96-well adapter that can fit into the four parts of the HTCP that allows multi-screening to be conducted with a microscope, enabling simultaneous observation of both cell types under different conditions and high-resolution imaging (Figure 2d).

The two containers, Body A and Body B were tightly coupled via the insert guide to prevent leakage. The claw in Body B is structured to constrict Body A tightly past the convex portion of the container (Figure 3a). The O-ring sandwiched between the containers at the junction of Body A, and Body B (Figure 3b) can prevent leakage even when a filter is not used.

The HTCP can be used in two ways (Appendix A): the separated culture method (A), where Body A and B are used independently and are only combined during the co-culture step, and the first assembly process (B), in which the state of the co-culture is regulated by controlling the volume of the culture solution. The container has a structural limitation so that co-culturing cannot occur if less than 400 μL of culture solution is used, meaning that the co-culture state can be achieved at any time by increasing the volume of culture liquid.

The use of a filter allows the liquid factors to move without mixing the cells; however, the nature of the filter may cause problems such as adsorption of the liquid factor. Therefore, we conducted initial experiments without a filter to confirm the subsequent effects of absorption by the filter (Appendix A). As the bottom of the HTCP is not composed of the filter material, three-dimensional culturing is also possible, along with the use of gel and coating (Appendix A). One of the essential advantages of this design is that because the filter is located away from the cells, the pores in the filter are not likely to be obstructed by cells, even at high cell densities.

### 3.2. Experimental Design

#### The Experimental Design for Evaluating HTCP

There are no previous studies on the rate at which metabolites secreted by cells move horizontally between vessels. The soluble substances secreted by the cells in the medium are thought to move between vessels via diffusion. Extracellular vesicles are also thought to move between vessels via Brownian motion due to their size. However, depending on the cellular factors, the actual situation may differ. The passing ratio of metabolites through the filter was not clear. The amounts of extracellular vesicles passing through are also unclear. Furthermore, it is unclear whether they move between vessels, in theory, even in the presence of cells.

We believe that it will be useful to evaluate the substance transfer properties using actual cells and HTCP for advancing the micromachine technology. Therefore, we devised an experimental design, shown in Appendix A, as an overview. First, we measured the migration properties of substances and extracellular vesicles through the filter without cells (Figure 4). Second, the concentration and mass transfer produced by the cells were evaluated (Figure 5). Third, the filter permeability of exosomes was investigated (Figure 6). Fourth, we evaluated the differences between VTCPs and HTCPs in terms of extracellular vesicle transfer (Figure 7). Finally, microscopic observation of exosomes passing through the filter and being taken up by neighboring cells was performed. Taking an advantage of the characteristics of the three-connected vessel, images were obtained showing that exosomes taken up from cells in both the left and right vessels were fused and processed together.

Here, we have provided an in vitro experimental system using a combination of HTCP and a track-etched membrane. Filters with pore sizes of 0.6 μm and 0.03 μm were mainly used to evaluate the mass transfer properties. Details of the filters are listed in Appendix A.

### 3.3. Measurement Results

#### 3.3.1. Differences in Mass Transfer Due to Differences in Wells Containing the Materials

We measured the change in the concentration of glucose to understand the relationships among the position, filter, and substance. Results showed that the glucose took a considerably longer time to pass through the 0.03 μm filter (Figure 5).

On the other hand, even the 0.03 μm filter had good permeability for NH_4_^+^ (Appendix A). The difference in time required for the metabolites to pass through was related to their sizes. These measurement results follow the theory of mass movement, known as the Brownian motion. The correlation approximation equations and correlation coefficients are provided in Appendix A. As the standard error is small and cannot be seen in the graph, the data are provided in Appendix A and an Excel file is available from MDPI’s online.

#### 3.3.2. Differences in Mass Transfer Due to Differences in Wells in Which the Cells Were Placed

Cells produce lactate and NH_4_^+^. Therefore, we investigated the production of lactate and NH_4_^+^ (Figure 5) in the culture medium. The increasing curves for lactate and ammonia both followed the polynomial approximation curve (Appendix A). In all vessels, the correlation coefficient was almost 1. These results indicate that the lactate and NH_4_^+^ secreted by the cells are always produced at a constant rate, and the transfer between vessels also follows the laws of physics.

#### 3.3.3. Validation of Protein Pass Rate

The “standard protein” is a mixture of proteins of different sizes, and the sizes are known. The standard protein was used to confirm the ability to pass through the filter. The presence and amount of protein passage were evaluated using the SDS-PAGE method and Coomassie protein assay. A standard protein was added to Body A (Appendix A). The amount of protein in Body B after 48 h was quantified by the Bradford method and confirmed by silver staining (Appendix A). Significant amounts of different proteins were observed to pass through the 0.6 μm filter, whereas barely any proteins passed through the 0.03 μm filter.

#### 3.3.4. Difference in Exosome Transfer Due to Difference in the Pore Size of the Filter

We examined the permeability of the HTCP for EVs using filters of two sizes: 0.6 and 0.03 μm. A filter with a hole diameter of 0.03 µm enables interrupting EVs from migrating to the neighboring vessels. A filter with a hole diameter of 0.6 µm enables the exosomes (a specific subgroup of EVs) to pass through the filter and migrate to the adjacent vessel. This was confirmed by Western blot analysis of the microparticles in the right container. The results showed no exosome surface markers (Figure 6b).

#### 3.3.5. Comparison of the VTCP and HTCP

As the HTCP was designed to analyze cell–cell communication via EVs, we next compared the rate of EVs passing through filters of the same size (1.0 μm). A track-etched membrane, which is a thin polymer film with geometrically well-defined and accurate pores (it4ip Inc.), was selected for this experiment. Approximately 1 × 10^4^ cells were seeded in one well, and the number of EVs in both vessels after 48 h was measured. The results showed that the HTCP contained two times the density of EVs per cell compared to the VTCP at 72 h, demonstrating the high efficiency of the HTCP for EV passage (Figure 7b).

#### 3.3.6. Confirmation of EVs Migration and Cellular Uptake

One of the advantages of the proposed device is that it can be used for biological cell culture and microscopic observation. Therefore, we used cells expressing CD63 protein fused to fluorescent proteins to confirm the migration of exosomes and their uptake by the cells. CD63 is an exosome surface marker. Plasmid transfection established stable pancreatic cancer cell lines expressing CD63-GFP (gPANC1) or CD63-RFP (rPANC1). gPANC1 and PANC1 cells were seeded in bodies A and B, respectively. GFP was observed in cells in Body B after 48 h, indicating that exosomes migrated to the adjacent cell from the Body A vessel (Figure 8a). Next, to take advantage of the characteristics of the three connected containers, we spread exosome-producing cells labeled with different colors in the containers at both the ends (gPANC1 and rPANC1). After 48 h, the center container was photographed (Figure 8b). The yellow color indicates that the cells may have processed exogenous exosomes at the same time. The biological significance of this experiment is that it can reveal the processing of the exosomes taken up by cells. However, as a limitation of the research, these images alone cannot be proven as significant due to optical resolution issues. Apart from the point of optical resolution, it has not been possible to conduct experiments to determine whether material from the left or right cell is preferentially processed under the same culture conditions. Changing the left and right cells to different types makes it possible to check whether the central cell prioritizes substances from either side.

## 4. Discussion

Originally, our aim was to investigate the interactions that occur between cancer cells and normal cells. We believe that the results of this paper will be useful not only for biological researchers who use HTCPs but also for future engineering researchers who develop microfluidic devices.

Microfluidic and co-culture systems have a high affinity. The co-culture method has been classically used for studying cell–cell interactions, either with a mixture of cells [23,24] or by adding the supernatant of one cell type to the culture of another cell type [25,26]. A co-culture vessel with upper and lower chambers is frequently used to prevent cells from mixing [27] while allowing examination of the impact that groups of cells have on each other.

However, various new cell culture methods have been proposed in recent years to overcome the limitations of these traditional approaches, which are summarized in Appendix A (modified and permitted from Shimasaki et al.) [1].

Co-culture methods can be classified according to the presence or absence of separation material and structural characteristics. The traditional and conventional methods use a filter and upper and lower chambers, forming a bucket-in-a-bucket structure, resulting in a vertical configuration (VTCP). Even today, VTCPs are widely used for co-culture. However, as no other effective co-culture vessels have been developed, no comparison has been made in terms of their performance.

As summarized in Appendix A (modified from Shimasaki et al. [1] with permission), the major advantages of the HTCP include the ability to confirm the conditions of various cell types simultaneously, use the same material on both sides of the filter, use high-density co-cultures with no detrimental effects, and use multiple types of filters.

In this study, our experiments revealed that HTCP has a high co-culture efficiency. One reason for the excellent co-culture effect is the ability to control the volume ratio of the containers and the positional relationship between the vessels and the filter.

The structure of VTCP in which the “buckets” are stacked means that the volume ratio of the culture fluid does not become 1:1, which results in a large dilution effect. Achieving the same volume ratio is desirable for the effective co-culture of each cell type (Figure 9a). The VTCP systems also allow placing cells directly onto the filter (Figure 9b), meaning that high cell densities lead to pore blockage, preventing the exchange of material through the pores.

Therefore, the decrease in the co-culture effect that can occur in cell cultures over long periods is not visually apparent. Some researchers have argued that in VTCPs, substances that are secreted by the upper cells pass through the filter to affect the lower cells. However, this could not be tested because there was no container with which the efficiency could be compared. The development of the HTCP has allowed such verification to be performed. The better performance of the HTCP is not surprising because the problems associated with filter blockage at high cell densities are bound to have deleterious effects on the results obtained with VTCPs. As the substance moves to an equilibrium concentration, a 1:1 volume ratio between containers allows each substance to influence the other effectively. In Brownian motion, liquid factors such as EVs also pass through the filter away from the cell, meaning that it does not matter if the filter and cells are at a distance.

The track-etched membrane filter, which can be regarded as a microfluidic channel, is an important component. Both the VTCP and HTCP connect two vessels through track-etched membrane filters, which is the same mechanism as that used in microfluidic channels. Although the vessel is not a microstructure in terms of size, the mass transfer characteristics obtained in this study will be useful for other microstructures. We selected a 0.03 μm filter for restricting the passage of exosomes and a 0.6 μm filter for allowing the passage of exosomes. The experimental results showed that the size of the filter used could control the passage of exosomes. It was also found that small molecules such as glucose, lactic acid, and NH_4_^+^ can traverse the filer rapidly, even when pore sizes of 0.03 μm were used. The ammonia in the medium was also observed to migrate rapidly in the multi-coupled experiments, but the concentration gradient was maintained when ammonia was continuously secreted by the cells. This gradient means that the degree to which a cell affects other cells can be controlled by considering the location of the wells in a multiconnected experiment. In addition, as shown in the results for the 0.03 µm filter for KLM in Appendix A, the fact that it took 72 h to equilibrate substances of extremely small size such as NH_4_^+^ and that the results were the same in multiple experiments indicates that the filter sandwiching mechanism that uses the O-ring in HTCP works well and is made with precision. This can be assumed because any leakage from a source other than the nano-sized filter pores will allow micro materials such as NH_4_^+^ to reach equilibrium in a very short time.

As shown in Figure 10, when the cells were placed in the left container (Q), the concentration of the substance from the cells increased in Q. When the cells were placed in the middle container (U), the concentrations in both containers (T and V) were the same. When the cells were placed in the second container from the left (X), W = Y did not occur because Y was diluted by the influence of Z, resulting in W > Y > Z.

In the relationship between concentration and time, the concentration of each container was represented by an approximate curve, with a correlation coefficient of approximately 1. These results indicate that metabolites and exosomes were not secreted randomly. They also show that the secreted material undergoes Brownian motion and movement due to osmotic disparity, moving precisely through the pores of the filter to the neighboring vessel. The degree of influence can change depending on the position of the cells in the connected plates. These results may also be applicable to sensor technology via filters and microfluidic channels. It was very interesting that the concentration of extracellular vesicles also changed along the approximate curve. This means that extracellular vesicles migrate according to the theoretical concentration curve even when the vessels are separated and filtered. In other words, the combination of horizontal co-culture vessels and the track-etched membrane made it possible to control the co-culture effect.

Co-culture is a powerful tool to observe the intrinsic cell–cell interaction of extracellular vesicles. To understand the reason for this, we need to know about exosome extraction techniques and research methods.

Several techniques have been proposed for the extraction of EVs [18,28,29] and the addition of collected EVs to other cells to confirm their effects. However, such artificial administration of EVs cannot be performed in vivo. Moreover, these extraction methods can only recover specific EV populations because of the nature of the approach. Techniques such as polymer-based precipitation [30], immunoaffinity capture using surface markers [31], purification by flow cytometry, and size-exclusion chromatography [32] have all been used to recover specific EVs. Among these methods, ultracentrifugation is the standard approach [33] because it can recover a particular population of EVs. However, the details of EVs have not yet been described, which means that experimental methods that can assess the precise nature and characteristics of exosomes are required. Indeed, information on the characteristics of exosomes remains largely unknown, although proteins and nucleic acids appear to be involved. Exosomes include features such as lipid bilayer membranes, tetraspanins such as CD9 and CD63, membrane transport proteins, and adhesion molecules such as integrins.

Similarly, when extracting exosomes, only the effects of a particular population can be examined. The addition of large numbers of EVs may be an artificial condition that would not otherwise occur, making it difficult to reproduce the natural phenomena that occur in cells. Although experiments in which EVs are extracted and administered are useful, this is also a one-sided experiment. Some phenomena occur only when the cells communicate with each other. Because a living cell has unique mechanisms that are akin to a personal identification number, only experiments that allow such interaction can decrypt a biological identification number. In this regard, the co-culture technique is useful in that it can reproduce natural phenomena without requiring the extraction of EVs. The use of co-culturing technologies is necessary to examine the essential interactions of EVs.

To date, many effects of EVs produced by cancer cells on normal cells, or vice versa, have been reported [26,34,35]. However, the simultaneous analysis of these effects has been limited owing to the lack of suitable techniques.

Using the HTCP experimental system, it is possible to conduct experiments with different degrees of influence from other cells in other wells in the same experiment. For example, to confirm the effect of cancer cells on normal cells, we can examine the difference between Y and Z in the experimental system, as shown in Appendix A. To compare the degree of influence exerted by the two different types of cells, the experiment shown in Appendix A is recommended. The difference in substance levels between ‘X’ and ‘Y’ occurred due to the difference in the degree of influence of ‘W’ and ‘Z.’

Thus, horizontal co-culture systems offer a useful tool to unravel the characteristics of cells. We believe that our newly designed HTCP system can help promote EV research methodologies and co-culture techniques. As shown in Figure 9 and Appendix A, HTCP can control the effects of co-culture by selecting the location of the cells.

## 5. Conclusions

The combined device of a vessel and a track-etched membrane can be considered as a microfluidic device. Vertical co-culture vessels, which have been conventionally used for a long time, are not sufficient for co-culture via liquid factors. In this study, we demonstrated a novel co-culture plate capable of controlling co-culture effects by clarifying the degree to which materials pass through a horizontal connection. The analysis showed that an approximate curve could represent the horizontal mass transfer rate with a high correlation coefficient; even if the cell and the channel are located away, the secretions of the cell will pass through the fluidic channel. We hope that our results will be useful for the design of other microfluidic devices.

For example, in the case of making a test kit using cells, the cells and fluidic channel may be designed to be close to each other. As the extracellular vesicles are secreted after the changes in the cell membrane, a situation where the cell membrane is in contact with the microfluidic channel may lead to failure in the secretion of extracellular vesicles, or the cell may block the channel itself. As our data show, the distance between the cell and the microfluidic channel does not affect the transfer of extracellular vesicles; rather, it is necessary to avoid blocking the microfluidic channel. We think it is important to consider the positional relationship between the cells and the microfluidic channel.

Finally, we expect that this cell co-culture vessel will be useful for expanding research applications in various fields, especially for understanding EVs and their roles in cell biology.

## Figures and Tables

**Figure 1 micromachines-12-01431-f001:**
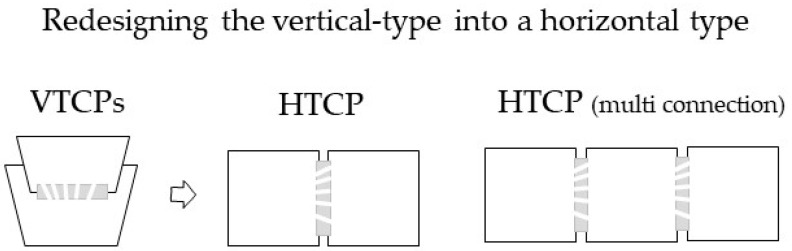
Redesigning the vertical-type co-culture plate to produce a horizontal-type co-culture plate (HTCP).

**Figure 2 micromachines-12-01431-f002:**
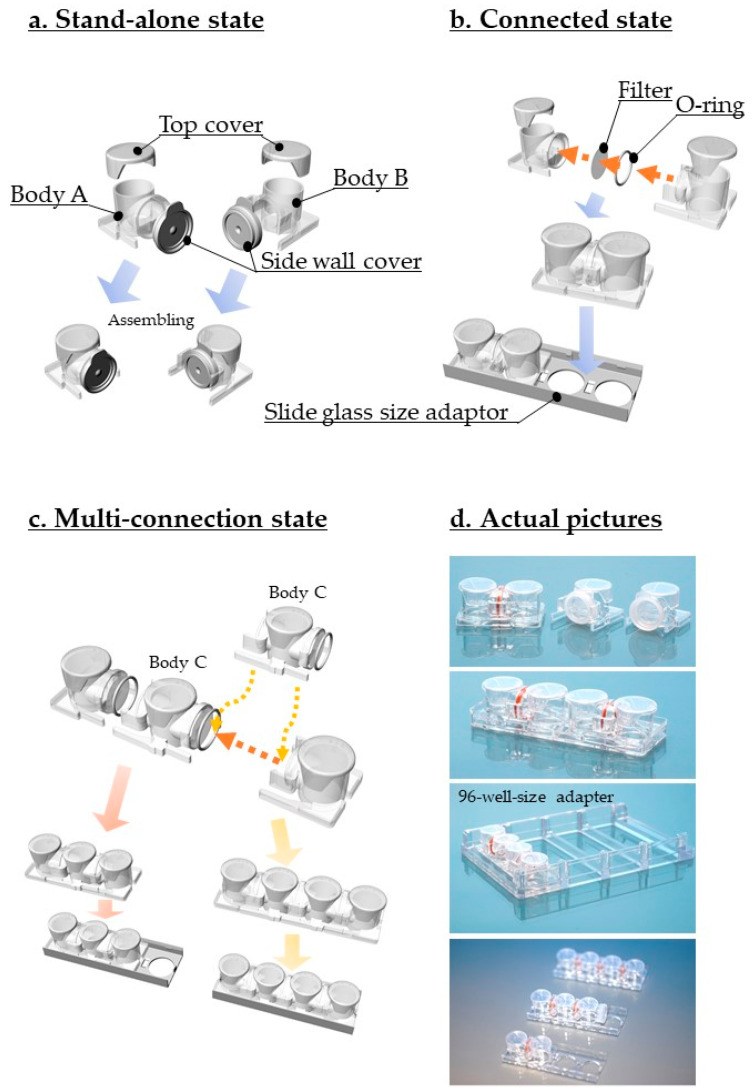
Design and description of the horizontal-type culture plate (HTCP). The culture vessel can be employed in two ways. (**a**) Initial monoculture mode. Bodies A and B are designed to allow the use of separated cell culturing. (**b**) Co-culture mode by connecting the two containers (combined state). In the connected mode, the two containers are joined from the beginning of cell culture. The height (volume) of the culture medium is used to control the amount of liquid that traverses the two containers, controlling the extent of co-culturing. (**c**) Body C is used for multiple connections, with a coupling mechanism on both the left and the right sides of the container. How to use are depicted in Appendix A. (**d**) Actual pictures.

**Figure 3 micromachines-12-01431-f003:**
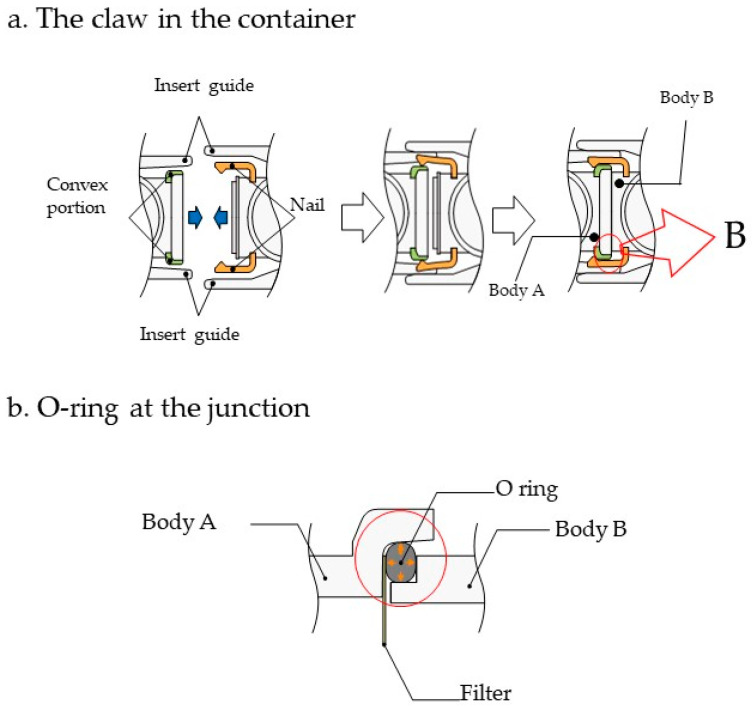
The mechanism used for leakage prevention and the methods used to prevent it. The containers (Body A and Body B) are tightly coupled via the insert guide to prevent leakage. (**a**) The claw in the container of Body B is structured to constrict Body A tightly past the convex portion of the container. (**b**) The O-ring at the junction of Body A and Body B prevents leakage.

**Figure 4 micromachines-12-01431-f004:**
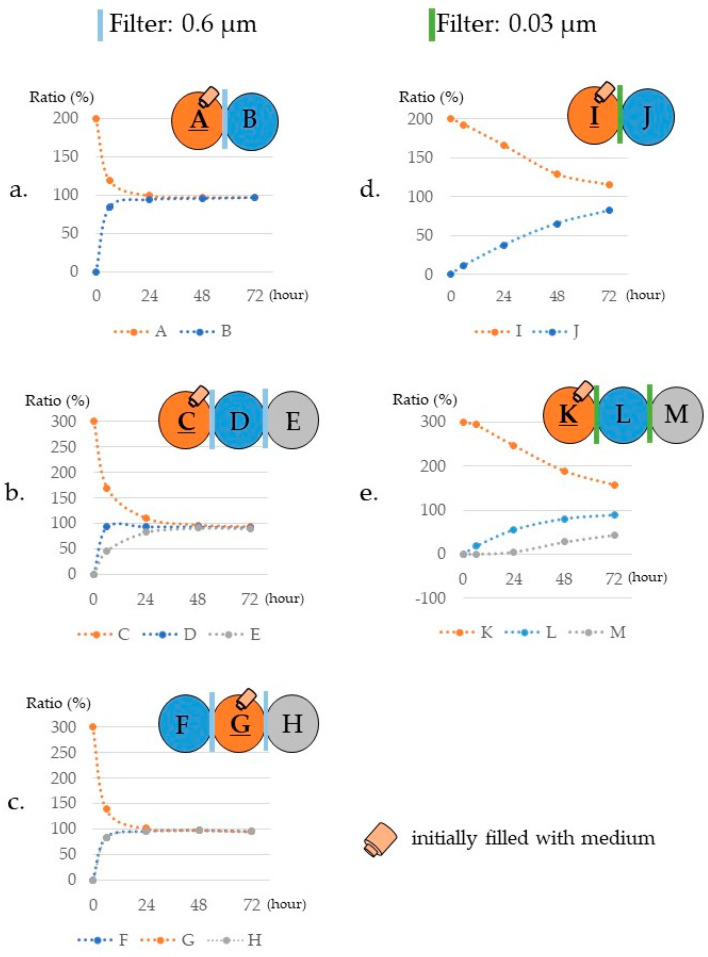
The results of glucose transfer without cells. Orange-colored wells with clip art of bottle were filled with medium including glucose. The remaining wells were filled with phosphate-buffered saline (PBS), which did not contain glucose. When all wells were mixed, the final concentration was converted to 100%. The vertical axis shows the converted concentration in %. Glucose concentration in the medium was measured after 6–72 h. The vertical axis is the concentration (%), and the horizontal axis is the time (h). The filter pore size is 0.6 μm (**a**–**c**) or 0.03 μm (**d**,**e**). (**a**) In case of 2 linkages and medium on the left side. (**b**) In case of 3 linkages and medium on the left side. (**c**) In case of three connections and a medium in the middle. (**d**) In case of 2 linkages and medium on the left side. (**e**) In case of 3 linkages and medium on the left side.

**Figure 5 micromachines-12-01431-f005:**
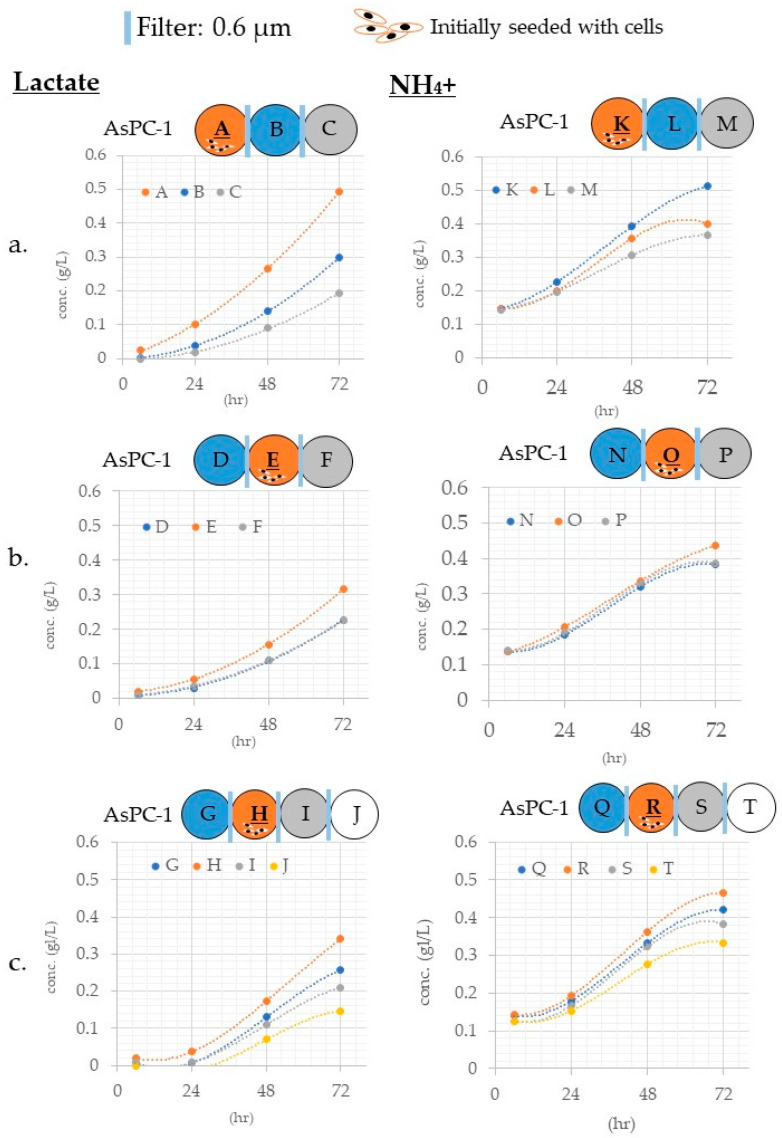
The same number of cells were placed in wells with clip art of the cell. Lactate and NH_4_^+^ concentration in the medium were measured regularly for 24–72 h. The vertical axis is the concentration, and the horizontal axis is the elapsed time. The results of the three or four connected plates are shown. Lactic acid measurement results on the left, NH_4_^+^ measurement results on the right. The polynomial approximation showed the best correlation. Correlation approximation equations and correlation coefficients are shown in Appendix A. (**a**) In case of 3 linkages and cells on the left side. (**b**) In case of 3 linkages and cells on the middle. (**c**) In case of 4 linkages and cells on the second from the left side.

**Figure 6 micromachines-12-01431-f006:**
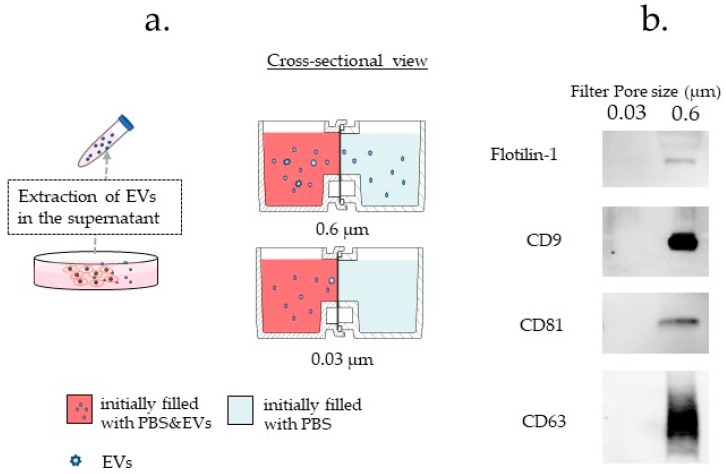
Validation of the ability of extracellular vesicles (EV) to pass through the filter according to pore size. (**a**) Scheme of the evaluation method for EVs secreted from PANC-1 cells. (**b**) Evaluation of the passing ability of extracellular vesicles (EVs). The same number of EVs from PANC-1 cells were seeded on one side (left side) of the horizontal-type co-culture plate (HTCP). The number of exosomes passing through the filter on the vessel’s right side were measured using Western blot. The 0.03 μm filter blocked the transfer of the EVs. The absence of the exosome on the adjacent vessel is confirmed by the Western blotting method in experiments with pore size 0.03 μm. Markers of exosomes (CD9, CD63, CD81, Flotilin-1) were not detected at 72 h.

**Figure 7 micromachines-12-01431-f007:**
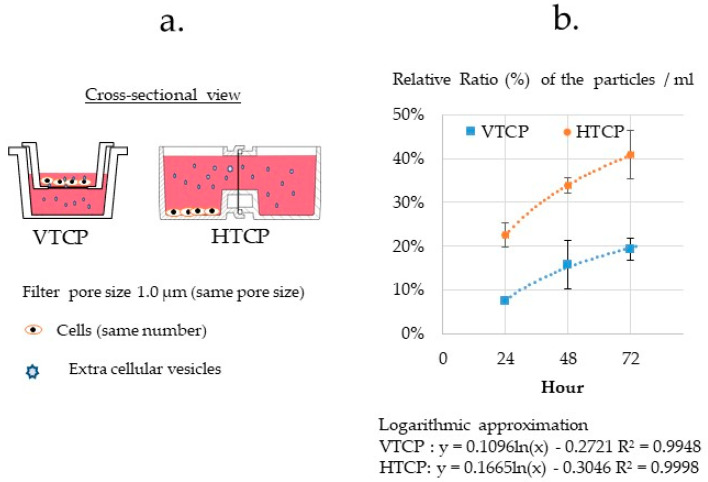
Validation of the ability of extracellular vesicles (EV) to pass through the filter according to co-culture type. (**a**) Scheme of the evaluation method for EVs secreted from PANC-1 cells. (**b**) Evaluation of the passing ability of extracellular vesicles (EV) secreted from cells. The same number of PANC-1 cells was seeded on one side of the horizontal-type co-culture plate (HTCP) and vertical-type co-culture plate (VTCP). The number of EVs passing through the filter was measured by Nanosight. The number of EVs per ml was then calculated. The EV density on the cell side was set to 100%. The HTCP contained three times the density of EVs per cell at 24 h, two times at 72 h, compared to the VTCP, demonstrating the high efficiency of the HTCP for EV passage.

**Figure 8 micromachines-12-01431-f008:**
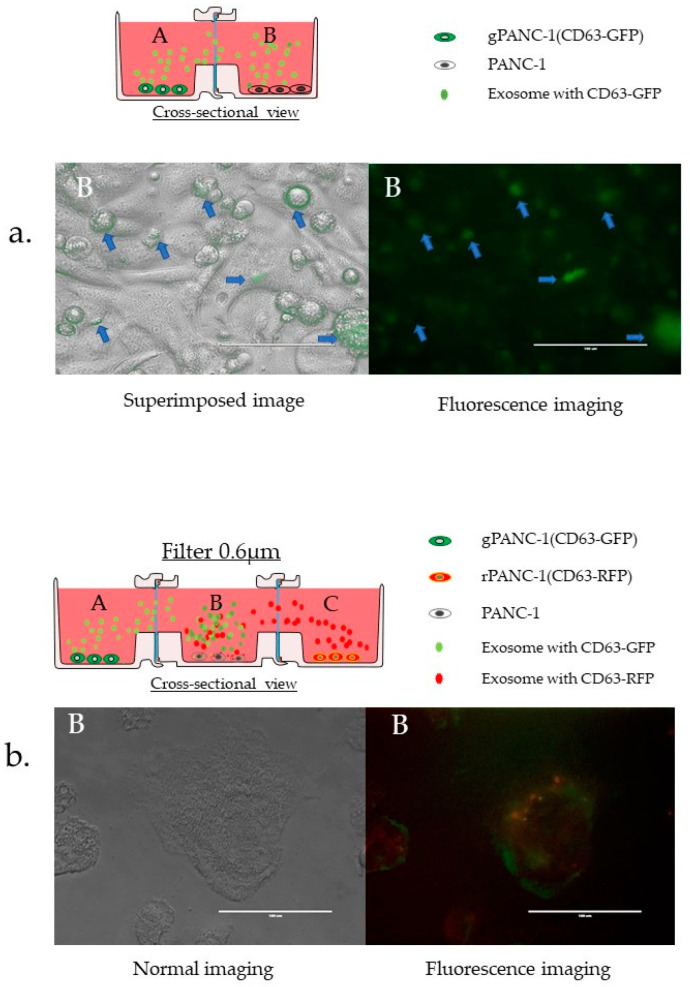
Confirmation of EVs migration and cellular uptake by microscope (**a**) Uptake of EVs by CD63-GFP-expressing cells. gPANC1 cells were seeded in Body A, and a photograph of Body B was taken 48 h later. (**b**) Uptake of extracellular vesicles from the both-side cells in the containers by three connected vessels. Unlabeled PANC-1 cells were seeded in Body C at the center, while gPANC1 and rPANC1 were seeded in Body A and B, respectively. Forty-eight hours later, mixed fluorescence was detected in the cells in Body C.

**Figure 9 micromachines-12-01431-f009:**
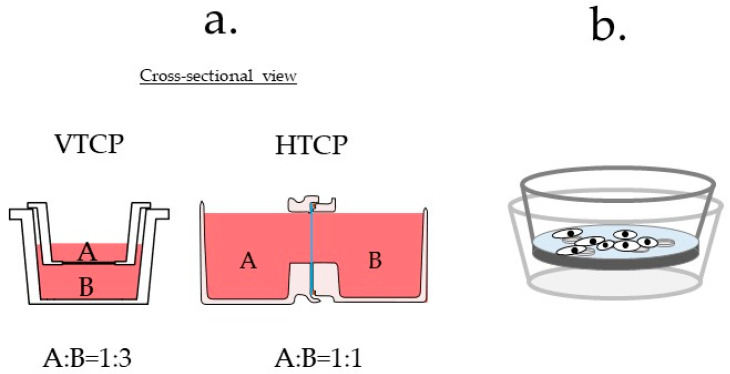
The reason why HTCP is more efficient in co-culture compared to VTCP. (**a**) The most efficient conditions are those in which the volume of the culture medium is the same. (**b**) As the cells are on top of the filter, the higher the cell density, lesser material can pass through the filter pores.

**Figure 10 micromachines-12-01431-f010:**
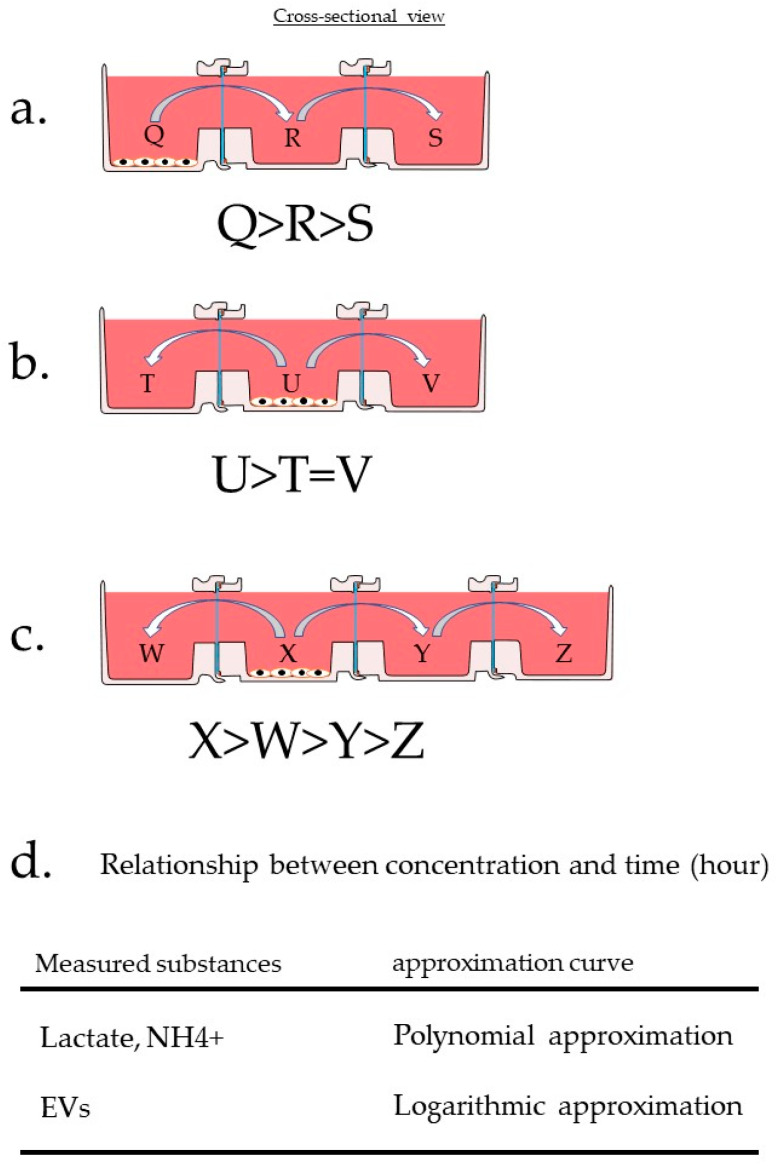
Material concentration gradient caused by multiple linkages (**a**) When cells are placed in the leftmost container (Q) in a three-container system, the concentration of the substances from the cells increases in Q. (**b**) When cells are placed in the middle container (U) of a 3-connected container, the concentrations of substances from the cells in the containers at both ends (T and V) are the same. (**c**) If the cells are placed in the second vessel from the left (X) in a four-connected container, W = Y will not be true because Y will be diluted by the influence of Z, resulting in W > Y > Z. (**d**) In the relationship between concentration and time, the concentration in each container can be represented by an approximate curve, with a correlation coefficient of almost 1. The approximate formulas and coefficients in each case are shown in the graphs. The data are described in Appendix A.

## Data Availability

The data that support the findings of this study are available from the corresponding author upon reasonable request.

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
