# Peer review of "Novel Platform for Regulation of Extracellular Vesicles and Metabolites Secretion from Cells Using a Multi-Linkable Horizontal Co-Culture Plate"

_micromachines, 2021, doi:10.3390/mi12111431_

Round 1

Reviewer 1 Report

I thank the authors for their submission.

All the figures in the manuscript needs cosmetic updates. 

The figures are leaking to the flowing pages, the fonts are different in most of the figures and the boxes drawn for each subfigures breaks the flow of the figure and the paper.

Since Figure 4 is first figure showing the design and the actual chips/platform developed for the study, I suggest authors to use more space and bigger pictures/drawings to show the design parameters of the chips/platforms.

The error bars in the figures are in the same colors, in black, for multiple parameters in the figures, using the same colors as the parameter will make easier to better understand to see which error bar belongs to which time point. 

Reviewer 2 Report

The paper is about a validation of horiozontal co-culture effects on cell-secreted exosome and metabolites. 

Although there are clear advantage of using the horizontal co-culture compared to vertical one, the article does not provide enough original contribution to the micromachine community. This is mainly due to the fact that commercial co-culture devices are used for the experiments. So I consider that the article could be rewritten for biology journals by focusing on original biological contributions. 

Major concern of this article is that it is not clear the original contribution of this article. I think it is mainly due to the fact that the present form of article focus too much on the validation of the author`s commercialized HTCP. This makes the article too long and blurs the core part of the contribution. This could readers to think simple validation and advertisement of commercialized product. So, I suggest that authors rewrite this article to be more concise to the point of the original contribution. If necessary, more parts which are not the core parts could be put in the appendix. Concerning the track-etched membranes, it is also not clearly presented if they are originally designed for specific problems. Authors need to clarify this point as well.

English should also be improved. 

Reviewer 3 Report

In this manuscript, the authors present the design and characterization of a co-culture apparatus by connecting two or more chambers horizontally. The communication of factors including EVs between the cell culture chambers can be regulated by using the filter of different pore sizes. Some points need further clarification.

Major concerns:

  • The material of the filter shall be mentioned.
  • Figure 7: it will be informative to show the size profile of EVs as well.
  • Section 3.2.5 and 3.2.6: the use of the term ”mobility” may be confusing. For example, mobility may mean electrical mobility for charged particles. The authors are suggested to define or replace the term “mobility”.
  • Line 631-632: the authors are suggested to rephrase the statement “it does not matter if the filter and cells are far apart”, since the distance affects the concentration profile and the time needed for diffusion. The authors are suggested to measure the profile and concentration of EVs over the elapsed time.
  • Can the cell chambers be separated again after assemble?

Minor concerns:

  • Abstract, line 36: please consider modifying “It is generally assumed is less …” to “It is generally assumed less …”.
  • Figure 1C: please consider rephrasing “exchange the exchange of materials”.
  • Section 2.9: it may not be straightforward to use a 20-ug aliquot of protein from a total EV protein of 2 ug.
  • Line 325: please consider modifying “silicon” to “silicone”.

Round 2

Reviewer 2 Report

The paper is revised based on the reviewer`s comments. Authors improved the presentation clarity by rewriting the article to be more close to the point. 

However I suggest that authors should further improve the presentation quality. 

  • The section 2 Materials and Methods should be developed more details. For example, the section 2.3 Microscopy has only single sentence. Authors need to details more in the section 2.
  • The proposed device is for biological cell culture but there is no microscopy images of showing the cell cultures. It is important to show them because microscopy is given as one of the advantage of using this technology. 
  • The section 5. Conclusion should also be more developed by giving a clear contributions of this paper. 
  • I suggest that authors can precise the view (top-view or side view) of the schematic images in the Figure 6,7 and 9.

Author Response

[2021/11/14]

Reviewer2

Special Issue Reviewer of Micromachines

MDPI AG, St. Alban-Anlage 66

4052 Basel, Switzerland

Tel. +41 61 683 77 34

Fax: +41 61 302 89 18

Dear Reviewer2:

Thank you very much for reviewing our manuscript, and we appreciate your comments and suggestions.

According to your suggestion and instruction, we are pleased to submit the revised manuscript entitled “Novel Platform for Regulation of Extracellular Vesicles and Metabolites Secretion from Cells using a Multi-linkable Horizontal Co-culture Plate” for publication in Micromachines.

We appreciate the positive comments and suggestions.

All changes to the manuscript are indicated in red font in the revised manuscript. For the revision, we have modified some figures and added a new figure in Figure 8. Therefore, the figure numbers are changed accordingly.

We hope that our revisions and responses described below adequately address all the comments from the reviewers. We believe that this paper conforms to the scope of micromachines and will be of great interest to its readers.

Sincerely,

Takeo Shimasaki, M.D., Ph.D.

Medical Research Institute

Kanazawa Medical University

Uchinada 1-1, Ishikawa 920-0293, Japan

Phone: 81-76-218-8312

FAX: 81-76-218-8312

Reviewer 2 comments

The paper is revised based on the reviewer`s comments. Authors improved the presentation clarity by rewriting the article to be more close to the point. However I suggest that authors should further improve the presentation quality. The section 2 Materials and Methods should be developed more details. For example, the section 2.3 Microscopy has only single sentence. Authors need to details more in the section 2. The proposed device is for biological cell culture but there is no microscopy images of showing the cell cultures. It is important to show them because microscopy is given as one of the advantage of using this technology. The section 5. Conclusion should also be more developed by giving a clear contributions of this paper.

I suggest that authors can precise the view (top-view or side view) of the schematic images in the Figure 6,7 and 9.

[Reply]

Thank you very much for reviewing our manuscript. We have added photomicrographs to Figure 8 and modified some figures. Therefore, the figure numbers are changed accordingly. The method and conclusion sections have been revised according to your suggestion. Product website information has been removed to avoid any misunderstanding (that it is intended for advertising purposes).

In the conclusion section, we have clearly expressed how this paper will contribute to micromachine researchers.

The title of the paper has also been revised to be more relevant to this conclusion.